# Controlling the Spread of Epidemics on Networks with Differential Privacy

**Dung Nguyen**
Department of Computer Science
Haverford College
Haverford, PA 19041
dnguyen1@haverford.edu

**Aravind Srinivasan**
Department of Computer Science
University of Maryland
College Park, MD 20742
srin01umd@gmail.com

**Renata Valieva**
Department of Mathematics
University of Maryland
College Park, MD 20742
rvalieva@umd.edu

**Anil Vullikanti**
Department of Computer Science
University of Virginia
Charlottesville, VA 22904
vsakumar@virginia.edu

**Jiayi Wu**
Department of Computer Science
University of Maryland
College Park, MD 20742
jwu12328@terpmail.umd.edu

## Abstract

Designing effective strategies for controlling epidemic spread by vaccination is an important question in epidemiology, especially in the early stages when vaccines are limited. This is a challenging question when the contact network is very heterogeneous, and strategies based on controlling network properties, such as the degree and spectral radius, have been shown to be effective. Implementation of such strategies requires detailed information on the contact structure, which might be sensitive in many applications. Our focus here is on choosing effective vaccination strategies when the edges are sensitive and differential privacy guarantees are needed. Our main contributions are $(\varepsilon, \delta)$-differentially private algorithms for designing vaccination strategies by reducing the maximum degree and spectral radius. Our key technique is a private algorithm for the multi-set multi-cover problem, which we use for controlling network properties. We evaluate privacy-utility tradeoffs of our algorithms on multiple synthetic and real-world networks, and show their effectiveness.

## 1 Introduction

A fundamental public health problem is to implement interventions such as vaccination to control the spread of an outbreak, e.g., [37, 8]. This is especially important in the early stages of an outbreak, when resources are limited. Here, we focus on network based models for epidemic spread, such as SI/SIS/SIR models, in which the disease spreads on a contact network $G = (V, E)$ from an infected node $u \in V$ to each susceptible neighbor $v$ of $u$ independently with some probability, e.g., [32, 1, 16, 36]; such models (which are simplifications of agent based models) have been used extensively in public health analyses in recent years. Interventions such as vaccination and isolation, can be modeled as node removal in such models [32]. The *Vaccination Problem* (VP), introduced

39th Conference on Neural Information Processing Systems (NeurIPS 2025).

in [15] for an SI type model (a similar version was considered in [21]), formalizes the design of an optimal vaccination strategy as choosing a subset $S \subset V$ so that the expected number of infections in the residual graph $G[V \setminus S]$ is minimized. This problem remains a challenging computational problem, and is NP-hard, in general [21, 15, 45].

Due to the computational hardness of the vaccination problem, a number of heuristics have been proposed for choosing a set $S$ to vaccinate, which involve choosing nodes based on properties related to the underlying contact network, such as degree and different notions of centrality, e.g., betweenness, pagerank and eigenscore [8, 9, 14, 16]; such heuristics have been shown to be much more effective that picking nodes randomly. In particular, choosing nodes which lead to a reduction in certain network properties of the residual network (i.e., after the vaccinated nodes are removed), below a critical threshold are quite effective. Examples of such strategies are reducing the maximum degree (the MAXDEG problem) [4, 36, 9], and the spectral radius (the MinSR problem) [46, 38]; we note that heuristics for the MAXDEG and MinSR problems have been used in many network based epidemic models, such as SIS, SIR, SEIR, etc. [38], as well as other contagion models, such as spread of influence [27, 43]. Optimal choice of such nodes (whose removal leads to the maximum reduction in such metrics) is also a difficult computational problem. There is a lot of work on approximation algorithms, e.g., [44, 46, 39, 41, 40], and is our focus here.

In most settings, data privacy is a fundamental challenge, due to the risk of revealing sensitive private information of users. For instance, individuals might wish to keep certain kinds of contacts private, since these might reflect sensitive activities they participate in. Privacy concerns were a major factor limiting user adoption of digital contact tracing apps [5]. Differential Privacy (DP) [13] has emerged as a very popular notion for supporting queries on private and sensitive data. Here, we study the problems of choosing nodes with edge DP guarantees to minimize the maximum degree (PRIVMAXDEG) and the spectral radius (PRIVMINSR); we note that the edge DP model has been studied quite extensively (the other commonly used model of node DP, e.g., [26, 49, 25], is not suitable for the PRIVMAXDEG and PRIVMINSR problems, since the goal is to output selected nodes). There has been recent work on different kinds of epidemic analyses with privacy, e.g., [7, 31, 30], and for more general problems of network science and graph mining, e.g., [34, 12, 10, 26, 49, 25]. However, *the* PRIVMAXDEG *and* PRIVMINSR *problems have not been studied so far*.

The PRIVMAXDEG and PRIVMINSR problems are closely related to a fundamental problem in combinatorial optimization, namely multi-set multi-cover. While there has been some work on covering problems with privacy, e.g., [19, 30, 11, 18], the version we study has not been considered before. Further, most of the prior work on covering problems with privacy, except [30], considers an *implicit* or *blackboard* model, which does not make the solution explicit; instead, sets which are part of the solution know this implicitly. This kind of implicit solution is not suitable for problems of epidemic control we consider here, and we design techniques to make our private solutions explicit. Our main contributions are summarized below.

**1. Minimizing the maximum degree with edge DP (PRIVATEMAXDEG).** We design Algorithm 1 (Section 4.2) for this problem, and show that it gives an $O(\ln n \ln(e/\delta)/\epsilon)$-approximation, with high probability. PRIVMAXDEG can be reduced to the private multi-set multi-cover problem (PRIVATEMULSET), a generalization of the set cover problem with privacy, which has not been considered before. We show that the iterative exponential mechanism can be used for PRIVATEMULSET (Section 4), and discuss how PRIVMAXDEG can be solved by reducing to it (Algorithm 1). We also show how to construct explicit solutions for PRIVMAXDEG (Algorithm 2) using the sparse vector technique [13].

**2. Minimizing the spectral radius with edge DP (PRIVMINSR).** This turns out to be a much harder problem because non-private algorithms use metrics (e.g., number of walks through a node) which have high sensitivity [44]. While the spectral radius satisfies $\rho(G) \leq \Delta$, where $\rho(G)$ and $\Delta$ denote the spectral radius and maximum degree, respectively, this bound can be quite weak in many graphs. We present two algorithms which lead to stronger bounds on $\rho(G)$ under different regimes (Section 5); the first is based on reducing the number of walks of a certain length, as in [44], and the second is in terms of the average degree of neighbors [17].

**3. Lower bounds.** It is well-known that for the covering problems, no differentially private algorithms can both output a non-trivial explicit solution and satisfy the covering requirement at the same time. We derive the lower bounds for even outputting an explicit partial coverage requirement, stating that any $(\epsilon, \delta)$-differentially private algorithm using no more than $O(\log n) + |OPT|$ must incur an

additive partial coverage requirement error of at least $\Omega(\log n)$. Similarly, for the PRIVATEMAXDEG, the explicit solution must have an additive error of at least $\Omega(\log n)$ for the target maximum degree.

**4. Experimental results.** We evaluate our methods on realistic and random networks. Our solutions lead to good bounds on both the maximum degree and the spectral radius. We find that implicit solutions have a higher cost relative to the non-private solutions, while the explicit solutions are quite sensitive to the privacy parameters, highlighting the need for carefully choosing the privacy parameters. We observe that our empirical results for the PRIVATEMAXDEG problem are consistent with the theoretical bounds we prove for our algorithms.

Most of the technical discussions — including algorithms, proofs, and experimental results — are deferred to the supplementary appendices due to space constraints. In particular, Appendices C.1.1 and C.1.2 provide full technical treatments of the Unweighted and Weighted variants of PRI-VATEMULSET, including the utility and runtime analyses of Algorithms 5 and 6, respectively. Additional technical details on the PRIVATEMAXDEG problem are presented in Appendix C.2, Appendix C.3 discusses lower bounds, while Appendix D contains the proofs and runtime analysis relevant to the PRIVMINSR problem. Finally, Appendix E contains additional experimental results.

## 2   Related Work

As mentioned earlier, the PRIVMAXDEG and PRIVMINSR problems have not been studied earlier. We briefly summarize prior work on two areas directly related to our work: (1) network-based epidemic control and (2) differential privacy for network and graph problems; additional discussion is presented in Section A in the appendix. There has been a lot of work on non-private algorithms for controlling epidemic spread on networks, e.g., [48, 14, 9, 33]. As mentioned earlier, strategies based on degree or centrality, e.g., [9, 33], have been shown to be quite effective in many classes of networks (including random graphs). There has also been prior work on reducing the spectral radius of the contact network, e.g., [41, 39, 40, 44, 35], which is closely related to the concept of epidemic threshold–a quantity that determines if there will be a large outbreak or not.

While there is a lot of work on private computation of different kinds graph properties (e.g., degree distribution, subgraph counts and community detection), e.g., [26, 22, 3, 23, 49], there is no prior work on the problems of controlling metrics related to epidemic spread. The most relevant work involves private algorithms for other problems in computational epidemiology , e.g., computing the reproductive number [7], estimation of the number of infections [31], and determining facility locations for vaccine distribution [30]. However, none of these methods imply solutions for the problems we study here.

## 3   Preliminaries

**Definition 3.1.** *A mechanism $M : \mathcal{X} \to \mathcal{Y}$ is $(\epsilon, \delta)$-differentially private if for any two neighboring inputs $X_1 \sim X_2$, and any measurable subset of the output space $S \subseteq \mathcal{Y}$, the following holds: $\Pr[M(X_1) \in S] \leq e^\epsilon \Pr[M(X_2) \in S] + \delta$ [13].*

When $\delta = 0$, we say that $M$ is $\epsilon$-differentially private. We study graph datasets, i.e., $\mathcal{X}$ corresponds to the set graphs with $n$ nodes. We consider the edge-DP model, where $V$, the set of nodes, is public and $E$, the set of edges, is kept private. More formally, two networks $G_1 = (V_1, E_1), G_2 = (V_2, E_2)$, are considered neighbors if $V_1 = V_2$ and there exists an edge $e$ such that $E_1 = E_2 \cup \{e\}$ or $E_2 = E_1 \cup \{e\}$ (i.e. they differ in the existence of a single edge). We note that there are other models of privacy in graphs, such as node DP, e.g., [26]; since our problems involve choosing subsets of nodes to be vaccinated, this model is not relevant here, and we only focus on edge DP.

We also utilize some standard privacy techniques and notations, such as the Exponential mechanism, Laplace mechanism, and AboveThreshold. See Appendix B for their definitions.

### 3.1   Problem Formulations

We study interventions for epidemic control, such as vaccination or isolation, which can be modeled as removing nodes from a contact network $G = (V, E)$ under the SIR model [32, 1, 16, 36]. Reducing

structural properties of the contact network—such as the maximum degree $\Delta(G)$ or the spectral radius $\rho(G)$ – can help limit epidemic spread [32, 38].

Let $n = |V|$ and $m = |E|$. For a graph $G$, let $d(v, G)$ denote the degree of a node $v$, and let $\Delta(G) = \max_v d(v, G)$ be the maximum degree in $G$. Let $\rho(G)$ denote the largest eigenvalue of the adjacency matrix of $G$. We also consider weighted graphs where $w(v)$ is the weight of node $v$.

**Definition 3.2.** *(PRIVMAXDEG problem) Given a graph $G = (V, E)$, a target max degree $D < \Delta(G)$, and privacy parameters $\epsilon, \delta$, the goal is to compute the smallest subset $S \subseteq V$ to remove (or vaccinate), such that the induced subgraph $G' = G[V \setminus S]$ satisfies $\Delta(G') \leq D$, while satisfying edge-DP.*

We refer to the non-private version of this problem as MAXDEG, and use $OPT_{\text{MAXDEG}}(G, D) = \min\{|S| : S \subseteq V, \Delta(G[V \setminus S]) \leq D\}$ to denote the optimal solution of the non-private version. Additionally, $GS_{\text{MAXDEG}}$ denotes the global sensitivity of MAXDEG.

**Definition 3.3.** *(PRIVMINSR problem) Given a graph $G$, a target threshold $\tau$, and privacy parameters $\epsilon, \delta$, the goal is to compute the smallest subset $S \subseteq V$ to remove, such that $\rho(G[V \setminus S]) \leq \tau$, while satisfying edge-DP.*

We refer to the non-private version of this problem by MinSR. Many bounds are known for the spectral radius, including: $\rho(G) \leq \Delta(G)$ and $\rho(G) \leq \max_v \sqrt{d(v, G)d_2(v, G)}$, where $d_2(v, G) = \sum_{u \sim v} d(u, G)/d(v, G)$ [24].

**Explicit and implicit solutions.** For the problems discussed above, the *explicit* version outputs an actual solution $S$ that satisfies edge-DP. However, for covering type problems, this is often challenging under DP [19]. We therefore also consider *implicit* solutions – these output a differentially private quantity $\pi$ such that each node $v$ can determine whether it is part of the solution based on $\pi$ and $G$.

## 3.2 Multi-set Multi-cover problem

To solve some of the above problems, we reduce them to the Multi-set Multi-cover problem, which we formally define as follows:

**Definition 3.4.** *(MULSET problem) Let $U = \{e_1, \ldots, e_n\}$ be a universe set on $n$ distinct elements. For each element $e \in U$, let the covering requirement $r_e$ be the minimum number of times $e$ must be covered, and let $R = \{r_e\}_{e \in U}$. Let $\mathcal{S} = \{S_1, \ldots, S_m\}$ be a collection of multi-sets, where each set $S_i$ contains $m(S_i, e)$ copies of element $e$. We refer to $m(S_i, e)$ as the multiplicity of $e$ in $S_i$. It is common to denote as $q = \max_{S \in \mathcal{S}} |S|$ the largest set size, and as $f = \overline{\max_{e \in U} |\{S \in \mathcal{S} | m(S, e) \geq 1\}|}$ the largest frequency of any element. The $\text{MULSET}(U, \mathcal{S}, R)$ asks to find the smallest sub-collection $\mathcal{S}' \subseteq \mathcal{S}$ such that each element $e$ in $U$ is covered at least $r_e$ times by the sets in $\mathcal{S}'$.*

*In the WEIGHTEDMULSET problem $(U, \mathcal{S}, R, C)$, each set $S \in \mathcal{S}$ has a cost, given by the function $C : \mathcal{S} \to \mathbb{R}$. The objective is to find a cover $\mathcal{S}'$ that minimizes the total cost, i.e., $\sum_{S \in \mathcal{S}'} C(S)$.*

Now, we consider the differentially private version of this problem, denoted PRIVATEMULSET. To match the edge-DP model described earlier, we define neighboring instances of the Multi-set Multi-cover problem as follows. Two instances $(U, \mathcal{S}, R)$ and $(U, \mathcal{S}', R')$ are said to be *neighbors* if one of the following conditions holds:

- There exists an element $e \in U$ such that $|r_e - r'_e| = 1$, and all other coverage requirements and sets are identical. That is, $\mathcal{S} = \mathcal{S}'$ and $R \triangle R' = \{r_e, r'_e\}$ for some $e \in U$.
- There exists an element $e \in U$ and an index $i \in [m]$ such that the multi-sets $S_i$ and $S'_i$ differ only in the multiplicity of $e$: $|m(S_i, e) - m(S'_i, e)| = 1$. All other sets and coverage requirements remain unchanged, i.e., $\mathcal{S} \triangle \mathcal{S}' = \{S_i, S'_i\}$ and $R = R'$.

Reducing a graph's degree-based objective – such as $\max_v d(v, G)$ or $\max_v d(v, G) \cdot d_2(v, G)$ – below a target threshold $D$ can be naturally formulated as an instance of the MULSET problem. Specifically, we define the universe as $U = V(G)$ and associate each vertex $u \in V(G)$ with a multi-set $S_u$ containing $u$ and its neighbors. The covering requirements $R$ are then defined to reflect how much the degree-related quantity, such as $r_v = \max(d(v, G) - D, 0)$ or $r_v = \max(d(v, G) \cdot d_2(v, G) - D, 0)$, must be reduced at each vertex. These reductions are described formally in the corresponding sections.

# 4 PRIVATEMULSET and PRIVATEMAXDEG Problems

We now describe private algorithms for reducing degree-based graph properties under the edge-DP model. These problems are reduced to instances of the PRIVATEMULSET framework introduced earlier. The intuition is the following: for example, in the MAXDEGREE problem, the utility of removing a node $v$ should naturally depend on how much its degree exceeds the threshold $D$, i.e., $\max(d(v, G) - D, 0)$. This translates naturally into the MULSET framework, where each element (e.g., an edge or neighborhood constraint) has a coverage requirement, and sets (vertices) contribute to meeting them. More generally, any problem where elements contribute toward satisfying some threshold-based constraints can be reduced to an instance of MULSET. We apply the same reduction principle to the SPECTRALRADIUS problem as well. In this section, we highlight the main ideas and results, using the private algorithm for MULSET as a black box. All formal details are deferred to the appendix.

## 4.1 Multi-set Multi-cover Problem: Algorithm and Analysis

In this section we discuss the **Unweighted** case. The algorithm and analysis of the **Weighted** case are similarly constructed, and are discussed in Appendix C.1.2. Our differentially private algorithm for the PRIVATEMULSET problem is inspired by [19]. The high-level idea is to assign a utility score to each set according to how much it contributes toward the remaining coverage requirements. The algorithm then repeatedly samples sets using the exponential mechanism – thereby ensuring differential privacy – based on these utility scores. Once there are no sets left, the algorithm outputs an *implicit* solution — a permutation $\pi \in \sigma(\mathcal{S})$ over the sets – rather than an explicit cover. The permutation defines a valid solution: for each element $e \in U$, we select the first few sets in the order of $\pi$ so that the coverage requirement $r_e$ is satisfied.

**Lemma 4.1.** *Algorithm 5 is $(\epsilon, \delta)$-differentially private, and its output is a solution to* PRI-VATEMULSET *of cost at most* $O((\ln m)/\epsilon + \ln q) \cdot |OPT|$ [1] *with probability at least* $1 - 1/m$, *where* $|OPT|$ *denotes the cost of an optimal non-private solution.*

## 4.2 The Private MaxDegree (PRIVATEMAXDEG) problem

The edge-privacy model of PRIVATEMAXDEG is equivalent to the privacy model of PRIVATEMULSET under a natural transformation described in Lines 1-6 of Algorithm 1. Specifically, a vertex $v$ with neighbors $\{u : u \sim v\}$ in PRIVATEMAXDEG corresponds to a star set in PRIVATEMULSET, so that the utility of selecting $v$ that is naturally proportional to its degree $\deg_v(G)$ is preserved in the PRIVATEMULSET instance as well.

For the privacy preservation, we observe that, under this transformation, neighboring graphs $G \sim G'$ that differ by a single edge $(u, v)$ map to PRIVATEMULSET instances that are at most 4-step neighbors. Specifically, the covering requirements for $u$ and $v$ change by at most 1 – i.e., $|r_u - r'_u| \leq 1$ and $|r_v - r'_v| \leq 1$ – and the multiplicities $m(S_u, v)$ and $m(S_v, u)$ also change by at most 1.

---

**Algorithm 1** Private algorithm for PRIVATEMAXDEG

---

1: **Input:** $(\epsilon, \delta)$, graph $G$, target degree $D$
2: Initialize set system $\mathcal{S} \leftarrow \emptyset$, requirements $R \leftarrow \emptyset$
3: **for** each $v \in V$ **do**
4:     Define multiset $S_v$ with $m(S_v, v) = \infty$ and $m(S_v, u) = 1$ for all $u \sim v$
5:     $\mathcal{S} \leftarrow \mathcal{S} \cup \{S_v\}, \quad R \leftarrow R \cup \{r_v = \max(\deg(v) - D, 0)\}$
6: **end for**
7: Set $\epsilon' \leftarrow \epsilon/4$, $\delta' \leftarrow \delta/4e^{3\epsilon'}$
8: **Return:** Algorithm 5$(\epsilon', \delta', \mathcal{S}, R)$ /*Applying the private multi-set algorithm*/

---

**Utility analysis.** Since we reduce the PRIVATEMAXDEG problem to an instance of the PRIVATEMULSET and apply Algorithm 5 to solve it, the utility of Algorithm 1 (stated by Theorem 4.2) follows the utility of Algorithm 5, by setting $m = |V|$, $q = 2GS_{\text{MAXDEG}} < 2|V|$ in Lemma 4.1. The

---

[1]Note that without the privacy constraints a greedy algorithm achieves an approximation ratio of $H(\max_S \sum_e m(S, e)) = O(\ln m)$, [42].

analysis for the weighted PRIVATEMAXDEG problem follows identically to the unweighted case, using Algorithm 6 for the weighted version of PRIVATEMULSET discussed earlier. Consequently, all arguments and results discussed previously are applicable with minimal modifications required for the utility bounds.

**Theorem 4.2.** *Algorithm 1 is $(\epsilon, \delta)$-differentially private, and the cost of its output is $\hat{B} < |OPT_{\text{MAXDEG}}| \cdot O((1 + 1/\epsilon') \ln |V|)$ with high probability.*

### 4.2.1 Explicit Solution

We also provide an explicit solution of which nodes to remove, incurring an additional privacy cost of $4\epsilon_1$. Unlike the implicit solution, the explicit output may allow some remaining nodes whose degrees exceed the the target degree threshold $D$. This approach builds upon the permutation $\pi$ generated by the implicit algorithm: by applying the AboveThreshold mechanism and removing only the first $k$ nodes in $\pi$, we ensure that the maximum degree is bounded by $D + O(\log m/\epsilon)$. The resulting solution removes at most $O(|\text{OPT}| \cdot \log k)$ nodes.

---

**Algorithm 2** Explicit solution algorithm for PRIVATEMAXDEG

---

1: **Input:** Instance of PRIVATEMAXDEG and a permutation $\pi$ obtained from the exponential mechanism
2: $T' \leftarrow 6 \ln n/\epsilon' - Lap(2/\epsilon_1)$
3: **for** $i = 1$ to $n$ **do**
4:    $\gamma_i \leftarrow L_i - Lap(4/\epsilon_1)$
5: **end for**
6: Let $k$ be the first index such that $\gamma_k \leq T'$
7: **Output:** $\{\pi(1), \ldots, \pi(k)\}$

---

**Utility Analysis.** Theorem 4.3 states the utility of the explicit solution output by Algorithm 2. We first observe that if we stop the algorithm at some iteration $\hat{k}$ where the selected node (and its equivalent set) no longer improves the coverage requirement by an amount $T = 6 \log n/\epsilon'$, then the maximum degree of the remaining graph is off from the target $D$ by an amount at most $O(\log n)$. Steps $2 - 6$ of the algorithm follows the AboveThreshold technique to select the first index $k$ that approximately satisfies the covering requirement of $\hat{k}$, i.e., the noisy utility $\gamma_k$ of the set chosen at step $k$ is relatively small enough (less than the noisy threshold $T'$ of the true target threshold $T$). We then utilize the accuracy guarantee of the AboveThreshold routine to argue that the selected index $k$ is in fact not too far away from the "true" stopping iteration $\hat{k}$ where its utility truly falls below the threshold $T$.

**Theorem 4.3.** *The output $k$ of Algorithm 2 satisfies $\Delta(G - \cup_{i=1}^{k}\{\pi_i\}) \leq D + O(\log n/\epsilon')$ with high probability. In addition, $k = O(OPT \cdot \log n/\epsilon')$ with high probability.*

**Lower bounds.** The explicit solutions cannot guarantee the coverage for the PRIVATEMULSET under DP guarantee. In this section, we argue that any explicit solution of the PRIVATEMULSET containing no more than $|OPT| + O(\log n)$ sets can only guarantees some partial covering with the additive error at least $\Omega(\log n)$. For the PRIVATEMAXDEG, the following lemma states the additive error of the target maximum degree, similar to the lower bounds of the PRIVATEMULSET, which we present in Appendix C.3.

**Lemma 4.4. Lower bound of PRIVATEMAXDEG.** *Any explicit $(\epsilon, \delta)$-differentially private algorithm for the PRIVATEMAXDEGREE removing at most $O(\log n) + |OPT|$ nodes with probability at least $1 - C, C = n^{-\Omega(1)}$, must incur an additive error $\Delta(G - \cup_{i=1}^{k}\{\pi_i\}) = D + \tilde{\Omega}(\log n)$, where $\pi(1), \ldots, \pi(k)$ are the removed nodes.*

## 5 Private SPECTRALRADIUS

In this section, we introduce two algorithms designed to reduce the spectral radius, $\rho(G)$, of a given graph $G = (V, E)$. In particular, these algorithms minimize specific graph metrics that upper bound $\rho(G)$.

## 5.1 Bound via PARTIALSETCOVER

Our first approach is based on the idea of reducing $|W_4(G)|$, the number of walks of length four, where $W_4(G)$ is the set of all such walks. We choose walks of length four because longer walks would have higher global sensitivity (generally, it is exponential in length of a walk), leading to larger privacy loss. Then, reducing $|W_4(G)|$ below $nT^4$ implies a bound on the spectral radius $\rho(G) \leq O(n^{1/4}T)$. Setting the threshold parameter $T = \Delta^{1/2}$ thus achieves a bound of $O(n^{1/4}\Delta^{1/2})$, which improves significantly over the bound $\rho(G) \leq \Delta$ when $\Delta = \Omega(\sqrt{n})$.

We employ the GREEDYWALK node selection algorithm from [44], which follows a greedy strategy to reduce the number of paths of a specified length. This algorithm, combined with the exponential mechanism, forms the first part of our approach. Further, we reduce our problem to an instance of the Partial Set Cover problem: each vertex in the graph corresponds to a set, and removing a vertex "hits" (or covers) a collection of walks that include it. Specifically, the utility of removing a vertex $v$ is defined as the number of walks of length $4$ that pass through $v$, formally given by: $A(v) = |\{w \in W_4(G) : v \in w\}|$. A differentially private algorithm for Partial Set Cover problem was introduced in [30], using an approach similar to Algorithm 3. However, it is important to note that in this context, the sensitivity of $|W_4(G)|$ is $\Delta^2$.

---

**Algorithm 3** Private Hitting Walks Algorithm for PRIVMINSR

---

1: **Input:** Graph $G = (V, E)$, privacy parameters $(\epsilon, \delta)$
2: Set $\epsilon' \leftarrow \epsilon/(2\ln(e/\delta))$, initialize permutation $\pi \leftarrow \emptyset$
3: **for** $i = 1$ to $n$ **do**
4:     Sample $v \in V$ with prob. $\propto \exp(\epsilon' \cdot A(v))$,    append $v$ to $\pi$ and remove $v$ from $V$
5: **end for**
6: Set $T \leftarrow \Delta^{1/2}, \theta \leftarrow 4nT^4, \hat{\theta} \leftarrow \theta - Lap(2/\epsilon_1)$
7: **for** $i = 1$ to $n$ **do**
8:     $\gamma_i \leftarrow W_4(G[V - \{\pi(1), \ldots, \pi(i)\}]) - Lap(4/\epsilon_1)$
9: **end for**
10: Let $k$ be the first iteration such that $\gamma_k \leq \hat{\theta}$
11: **Output:** $(\pi(1), \ldots, \pi(k))$

---

**Lemma 5.1.** *Algorithm 3 is* $(\Delta^2(\epsilon + \epsilon_1), \Delta^2\delta e^{(\Delta^2-1)\epsilon})$*-differentially private. If* $T^4 \geq 6\ln n/\epsilon'$*, the output* $V' = \{\pi_1, \ldots, \pi_k\}$ *of Algorithm 3 satisfies* $W_4(G[V \setminus V']) \leq nT^4 + O(\log n/\epsilon')$ *and gives an* $O(\log n)$ *approximation with high probability.*

## 5.2 Bound via PRIVATEMULSET

We can also apply Algorithm 5 for PRIVATEMULSET to indirectly reduce $\rho(G)$. According to [17], the spectral radius is bounded by $\rho(G) \leq \max_{u \in V(G)} \sqrt{\sum_{v \sim u} d(v, G)}$, which is always better than the trivial bound $\rho(G) \leq \Delta$. This improvement is especially significant in degree-disassortative graphs, where high-degree vertices are typically adjacent to many low-degree vertices. For further discussion see Appendix D.2.

---

**Algorithm 4** Private algorithm for PRIVATEMAXDEG

---

1: **Input:** $(\epsilon, \delta)$, input graph $G$, target max degree $D$
2: $\mathcal{S} = \{S_v : v \in V\}$, such that $S_v$ contains $\infty$ copies of $v$ and $d(u, G)$ copies of each $u$ that is adjacent to $v$
3: $R = \{r_V : v \in V\}, \forall v \in V : r_v \leftarrow \max(\sum_{u \sim v} d(u, G) - D, 0)$
4: $\epsilon' \leftarrow \epsilon/4\Delta$
5: $\delta' \leftarrow \delta/4\Delta e^{(4\Delta-1)\epsilon'}$
6: **Return** Algorithm 5($\epsilon', \delta', \mathcal{S}, R$)

---

**Theorem 5.2.** *Algorithm 4 is* $(\epsilon, \delta)$*-differentially private, and the cost of its output is* $\hat{B} < |OPT| \cdot O((1 + 1/\epsilon')\ln|V|)$ *with high probability.*

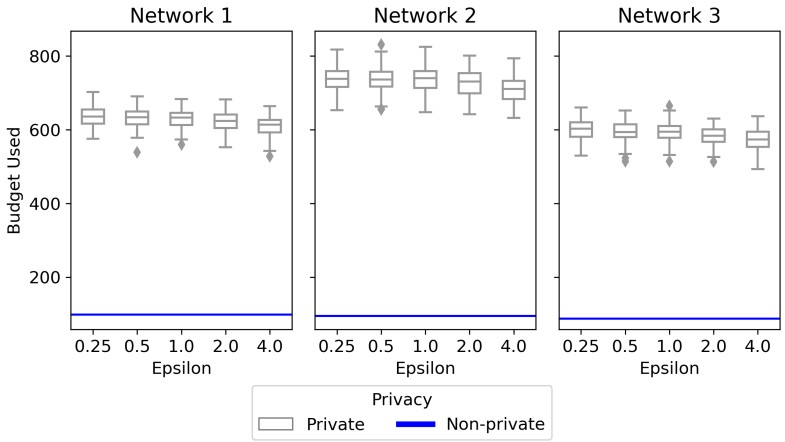

Figure 1: Effect of Privacy on Budget Requirements on Montgomery County Subnets

## 6 Experimental evaluation

We evaluate the performance of our algorithms on different realistic and random networks in terms of the following questions
• Effects of privacy budgets on the utility of our algorithm (both in terms of vaccination budget and epidemic metrics $\Delta(G)$ and $\rho(G)$).
• Tradeoff between vaccination cost, different epidemic metrics, and privacy parameters.
• Comparison between the implicit and explicit solutions.

| Graph Name | #nodes | #edges |
|---|---|---|
| Subgraph of digital twin of contact network for Montgomery, VA [16] | 10,000 | 83842, 84025, 84549 |
| BTER [28] with Power Law Degree ($\gamma = 0.5, \rho = 0.95, \eta = 0.05$) | 1000 | 31530, 31582, 31621 |

Table 1: Network datasets used in evaluation

**Datasets and setup.** We consider two classes of networks, as summarized in Table 1. The digital twin of a contact network [2, 16] is a model of real world activity based contact networks; we consider three subgraphs with 10,000 nodes of the network for Montgomery county VA. The BTER model [28] is a random graph model, which preserves both degree sequence and clustering; we consider three randomly generated networks. Both classes of networks have been used in a number of epidemiological analyses, e.g., [32, 8, 1].

**Effect of privacy on solution cost for the PRIVATEMAXDEG problem.** Figure 1 shows the cost of the implicit solutions computed using Algorithm 1 for the three subgraphs of the Montgomery county networks (labeled as Network 1-3). We use a privacy budget of $\delta = 10^{-6}$ and $\epsilon \in \{0.25, 0.5, 1, 2, 4\}$, and set a target degree of $D = 45$. For each $\epsilon$, we show a distribution over results computed by multiple runs of the algorithm. As described in the implicit Algorithm 5 for PRIVATEMULSET, the implicit solution is computed and plotted here. The cost of the solution to a non-private greedy algorithm for the multi-set multi-cover problem (which has a $H_\Delta$-approximation[42], where $H_n$ denotes the $n$-th harmonic number) is shown as the **baseline**. We note that the solution of Algorithm 5 is within a factor of about 10 of the non-private baseline, which could be viewed as being consistent with Theorem 4.2; further, the cost of the private solutions has a slight reduction with $\epsilon$.

Figure 2 shows the impact of privacy cost on the cost of the explicit solution for PRIVATEMAXDEG for the three BTER networks (Table 1) computed by Algorithm 2 with a target $D = 20$. We pick $\delta = 1/n = 10^{-3}$ here, and have relaxed the privacy to the multi-set multi-cover definition rather than the edge private definition of neighboring datasets. The results show, somewhat counter-intuitively, that the solution cost actually increases with $\epsilon$. Since in explicit solutions the solution cost is mainly

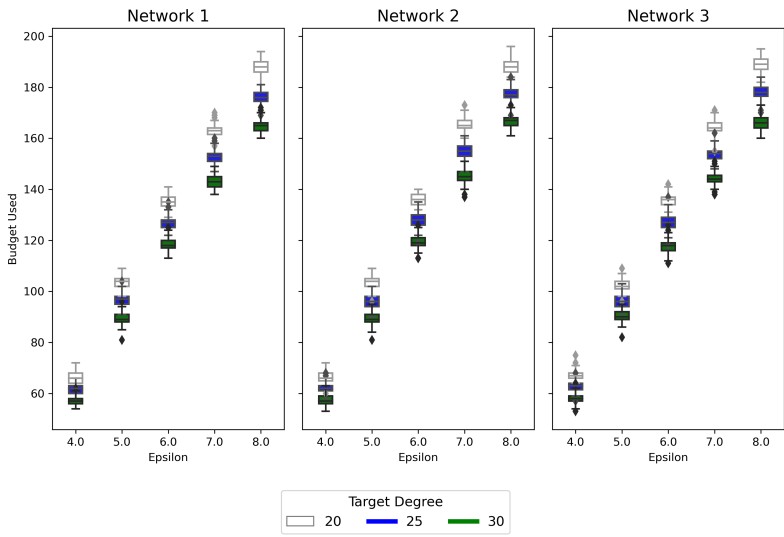

Figure 2: Effect of Privacy on Budget Requirements on BTER Graphs

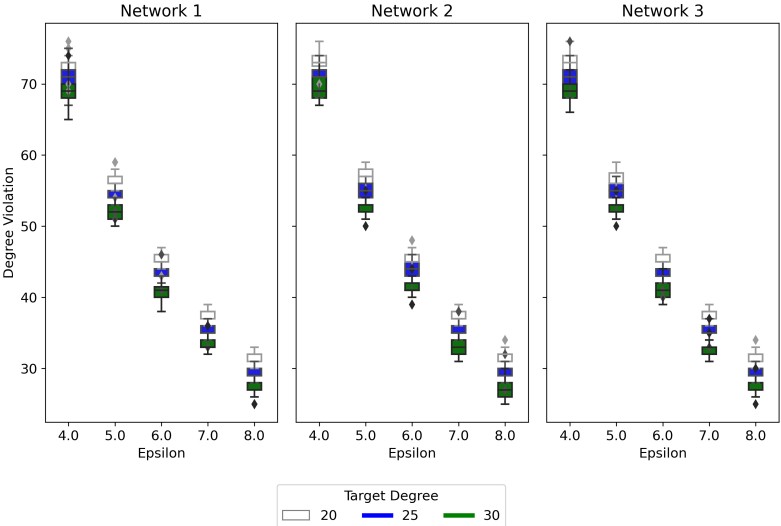

Figure 3: Effect of Privacy on Max Degree Violation on BTER Graphs

determined by Above Threshold step (in Algorithm 2), which allows lower $\epsilon$ to halt set selection earlier (before certain vertices meet their cover requirements), the algorithm is closer to fulfilling the entire covering requirement as in the non-private version as $\epsilon$ increases, which explains this behavior. This is also consistent with Figure 3, which shows that the resulting violation in the maximum degree from the target decreases significantly with $\epsilon$. This suggests that the choice of the privacy budget needs to be done carefully.

**Implicit vs Explicit solutions.** We investigate the performance of the implicit and explicit solutions (Table 2). The main difference between the two methods lies in when the permutations terminate, explicit would halt before the target degree is fully satisfied whereas implicit would not. This is demonstrated in that implicit solutions perform much better with metrics like max degree whereas explicit solutions have significantly lower vaccination costs.

Table 2: Comparison of Average Performance of Implicit vs Explicit Solutions ($\epsilon = 4.0$)

| $\gamma$ | $\rho$ | $\eta$ | EXPLICIT? | BUDGET | MAX DEGREE | SPECTRAL RADIUS |
|------|------|------|------|------|------|------|
| 0.3 | 0.95 | 0.05 | YES | 83.89 | 92.78 | 72.28 |
|  |  |  | NO | 506.62 | 20 | 18.35 |
| 0.5 | 0.95 | 0.05 | YES | 66.19 | 92.80 | 77.99 |
|  |  |  | NO | 430.36 | 20 | 18.55 |

## 7 Conclusion

We initiate the study of the challenging and largely unexplored problems of epidemic control on networks under differential privacy. Our focus is on the approach of removing nodes from a graph to optimize certain properties, such as the maximum degree and spectral radius of the residual graph, which models the vaccination effect on a contact network. We design the first set of algorithms along with rigorous utility analyses for minimizing the maximum degree and spectral radius under the edge differential privacy model. One of our main techniques involves transforming these problems into a multi-set multi-cover problem and using its private solution to determine the sets of nodes to be removed (or vaccinated). While providing explicit solutions for covering-type problems is challenging, we employ the sparse vector technique to relax the covering requirement, allowing for approximate explicit solutions that can be used to design vaccination strategies. The experimental results of our algorithms, evaluated on multiple realistic and random networks, demonstrate good privacy-utility trade-offs.

**Acknowledgments.** We thank the NeurIPS reviewers for their thoughtful comments.

**Disclosure of Funding.** This research is partially supported by NSF grants CCF-1918656, CNS-2317193, IIS-2331315, CCF-1918749 and CNS-2317194, and the Virginia Commonwealth Cyber Initiative Cybersecurity Research Award.

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

## Supplementary Material

## A  Related Work

We include additional discussion on related work here.

[7] studied the problem of estimating the reproductive number $R_0$ of an epidemic on its contact network in the SIS and SIR models. The reproductive number $R_0$ is closely related to the spectral radius, i.e., the reproductive number $R_0$ can be expressed as a function of the first eigenvalue of the adjacency matrix. Their privacy model protected the "weights" of the weighted contact network. Moreover, the work did not specify or imply any approach to modify the contact network to reduce such quantity, in order to reduce the spread of the pandemic.

There has been several work to calculate the spectral radius of an input graph–that is of independent interest from the perspective of epidemic control. [47] computed the eigenvalues and eigenvectors of an input graph under the edge-differential privacy. [6], also under the egde-differential privacy model, estimated the second smallest eigenvalues ($\lambda_2$), which is also commonly refered to as "algebraic connectivity". Similarly, [20] studied the same problem, but also considered the problem under the node-differential privacy model, in which two neighbor graphs differ by a node and its adjacent edges. None of the work suggested a method to reduce the spectral radius of the input network.

## B  Background

We briefly discuss the basic ideas of DP here; see [13] for more details.

**Definition B.1 (Exponential mechanism).** *Given a utility function $u : \mathcal{X}^n \times \mathcal{R} \to \mathbb{R}$, let $\mathrm{GS}_u = \max_{r \in \mathcal{R}} \max_{x \sim x'} |u(x, r) - u(x', r)|$ be the global sensitivity of $u$. The exponential mechanism $M(x, u, \mathcal{R})$ outputs an element $r \in \mathcal{R}$ with probability $\propto \exp(\frac{\epsilon u(x,r)}{2\,\mathrm{GS}_u})$.*

**Lemma B.2.** *The exponential mechanism is $\epsilon$-differentially private. Furthermore, for a fixed dataset $x \in \mathcal{X}^n$, let $OPT = \max_{r \in \mathcal{R}} u(x, r)$, then the exponential mechanism satisfies*

$$\Pr[u(x, M(x, u, \mathcal{R}) \leq OPT - \frac{2\,\mathrm{GS}_u}{\epsilon}(\ln |\mathcal{R}| + t)] \leq e^{-t}.$$

**Definition B.3 (Laplace mechanism).** *Let $f : \mathcal{X}^n \to \mathbb{R}^d$ be a function with global $\ell_1$-sensitivity $\Delta_f = \max_{x \sim x'} |f(x) - f(x')|_1$. The Laplace mechanism releases*

$$M(x) = f(x) + (Z_1, \ldots, Z_d),$$

*where $Z_i \sim \mathrm{Lap}(\Delta_f / \epsilon)$ are independent random variables drawn from the Laplace distribution with scale parameter $\Delta_f / \epsilon$.*

**Lemma B.4.** *The Laplace mechanism is $\epsilon$-differentially private. Moreover, each coordinate of the output is concentrated around the true value of $f(x)$, with noise magnitude proportional to $\Delta / \epsilon$.*

**Definition B.5 (AboveThreshold).** *Let $f_1, \ldots : \mathcal{X}^n \to \mathbb{R}$ be a sequence of queries with sensitivity 1. Given a threshold $\tau$ and a privacy parameter $\epsilon$, the AboveThreshold mechanism adds Laplace noise $\mathrm{Lap}(2/\epsilon)$ and $\mathrm{Lap}(4/\epsilon)$ to the threshold and to each query, and returns the first index $i$ such that the noisy query exceeds the noisy threshold.*

**Lemma B.6.** *The AboveThreshold mechanism is $(\epsilon, 0)$-differentially private.*

## C  PRIVATEMULSET and PRIVATEMAXDEG Problems

We start by describing and analyzing an algorithm for solving PRIVATEMULSET problem that is later used for PRIVATEMAXDEG.

### C.1  PRIVATEMULSET

#### C.1.1  Unweighted case.

We present an algorithm for the PRIVATEMULSET problem, building upon the framework and analysis from [19]. First, we define a utility function $A : \mathcal{S} \to \mathbb{R}_{\geq 0}$. For a set $S_i \in \mathcal{S}$ and an element

$e \in U$, the marginal utility is defined as $A(S_i, e) := \min(m(S_i, e), r_e)$, and then the total utility of a set $S_i$ is given by $A(S_i) = \sum_{e \in S_i} A(S_i, e)$. The algorithm will sample sets based on these utility scores computed in each iteration.

It is important to note that directly outputting an explicit solution – i.e., listing only the sets that form a valid cover – would violate differential privacy. In particular, this is because the solution to the vertex cover problem, which is a special case of the set cover problem, is known to retain privacy iff the output contains at least $|V| - 1$ vertices[19]. Therefore, in order to preserve privacy, the algorithm must produce an *implicit* solution, typically in the form of a permutation $\pi \in \sigma(\mathcal{S})$ over the sets in $\mathcal{S}$. Intuitively, $\pi$ should have the sets arranged in the order of decreasing utility. This ordering implicitly defines a cover: for each element $e \in U$, we select the first few sets in $\pi$ that would fully cover $e$. Formally, let $\pi_e := \{\pi(i) \mid 1 \le i \le n : \min(\sum_{j=1}^{i} m(S_{\pi(j)}, e), r_e) - \min(\sum_{j=1}^{i-1} m(S_{\pi(j)}, e), r_e) > 0\}$ be the indices of sets that contribute to covering $e$ according to $\pi$. Then $\{S_j : j \in \bigcup_{e \in U} \pi_e\}$ forms a valid multi-cover of $U$.

---

**Algorithm 5** Private algorithm for PRIVATEMULSET

---

1: **Input:** privacy parameters $(\epsilon, \delta)$, set system $\mathcal{S}$, covering requirement $R$
2: Set $\epsilon' \leftarrow \frac{\epsilon}{2\ln(e/\delta)}$
3: Initialize empty permutation $\pi \leftarrow \emptyset$
4: Initialize $r_e^{(0)} \leftarrow r_e$ for all $e \in U$, $\mathcal{S}^{(1)} \leftarrow \mathcal{S}$
5: **for** $i = 1$ to $|\mathcal{S}|$ **do**
6:     Define $A^{(i)}(S) := \sum_{e \in S} \min(m(S, e), r_e^{(i-1)})$
7:     Sample $S_j \in \mathcal{S}^{(i)}$ with probability $\propto \exp(\epsilon' A^{(i)}(S_j))$
8:     Append $j$ to $\pi$: $\pi(i) \leftarrow j$
9:     Update available set system: $\mathcal{S}^{(i+1)} \leftarrow \mathcal{S}^{(i)} \setminus \{S_j\}$
10:     **for** $e \in S_j$ **do**
11:         Update covering requirement: $r_e^{(i)} \leftarrow \max(0, r_e^{(i-1)} - m(S_j, e))$
12:     **end for**
13: **end for**
14: **Output** permutation $\pi$

---

**Lemma C.1.** *The output of the Algorithm 5 is at most $O(\ln m/\epsilon' + \ln q)OPT$ with probability at least $1 - 1/m$, where $OPT$ denotes the cost of an optimal non-private solution, and $q = \max_S \sum_e A(S, e)$ is the size of the largest set.*

*Proof.* Without loss of generality, we may assume that the permutation $\pi$ output by the Algorithm 5 is $\pi = (1, 2, \ldots, m)$. In other words, the sets $S_1, S_2, \ldots, S_m$ are sequentially added to the cover in this exact order.

Let $L^{(i)} := \max_{S \in \mathcal{S}^{(i)}} A^{(i)}(S) = \max_{j \ge i} A^{(i)}(S_j)$ be the maximum utility possible at step $i$; this also implies that there is a multi-set of that utility. Then the probability of selecting a set of small utility $< L^{(i)} - 3\frac{\ln m}{\epsilon'}$, given that we have at most $m$ sets to select from, is less than

$$\frac{m \cdot exp(\epsilon' L^{(i)} - 3\ln m)}{exp(\epsilon' L^{(i)}) + m \cdot exp(\epsilon' L^{(i)} - 3\ln m)} = \frac{1/m^2}{1 + 1/m^2} \le 1/m^2.$$

Next, consider two cases:

**$L^{(i)} > 6\frac{\ln m}{\epsilon'}$.** The probability that every multi-set selected has utility at least $L^{(i)} - 3\frac{\ln m}{\epsilon'} > L^{(i)}/2$ is $\ge (1 - 1/m^2)^m \ge (1 - 1/m)$. Because the greedy approximation is a $O(\ln q)$ approximation, Algorithm 5 can cover this region in at most $O(OPT \ln q)$ multi-sets with high probability.

**$L^{(i)} \le 6\frac{\ln m}{\epsilon'}$.** At this point there are at most $OPT \cdot L^{(i)}$ elements that require covering, and $OPT \cdot L^{(i)} \le OPT \cdot O(\frac{\ln m}{\epsilon'})$. Since the post-processing of the implicit solution selects only sets that cover at least one element, covering the remaining $O(OPT \frac{\ln m}{\epsilon'})$ elements takes an additional $O(OPT \ln m/\epsilon')$ sets.

Therefore, the algorithm uses at most $O(OPT(\ln m/\epsilon' + \ln q)$ sets. $\qquad\square$

**Lemma C.2.** *Algorithm 5 is $(\epsilon, \delta)$-DP.*

*Proof.* Similar to Lemma C.1, we assume that $\pi = (1, 2, \ldots, m)$, so that $\mathcal{S}^{(i)} = \{S_j : j \geq i\}$.

First, we consider neighboring problems $\mathcal{A} = (U, \mathcal{S}, R)$, $\mathcal{A}' = (U, \mathcal{S}', R)$ that share the same coverage requirements $R = R' = \{r_e : e \in U\}$ but differ in the multiplicity of a particular element $e_0$ in one set, such that $|m(S_k, e_0) - m(S'_k, e_0)| = 1$ for some $k \in [m]$. Define $t$ as the epoch when element $e_0$ is fully covered in both instances $\mathcal{A}$, $\mathcal{A}'$. We wish to establish a bound for $\frac{\Pr[M(\mathcal{A})=\pi]}{\Pr[M(\mathcal{A}')=\pi]}$:

$$\frac{\Pr[M(\mathcal{A}) = \pi]}{\Pr[M(\mathcal{A}') = \pi]} = \prod_{i=1}^{n}\left(\frac{e^{\epsilon' A^{(i)}(S_i)}}{\sum_{j=i}^{n} e^{\epsilon' A^{(i)}(S_j)}}\right) \Bigg/ \prod_{i=1}^{n}\left(\frac{e^{\epsilon' A^{(i)}(S'_i)}}{\sum_{j=i}^{n} e^{\epsilon' A^{(i)}(S'_j)}}\right)$$

$$= \frac{e^{\epsilon'(\sum_{i=1}^{n} A^{(i)}(S_i))}}{e^{\epsilon'(\sum_{i=1}^{n} A^{(i)}(S'_i))}} \cdot \prod_{i=1}^{n} \frac{\sum_{j\geq i} e^{\epsilon' A^{(i)}(S'_j)}}{\sum_{j\geq i} e^{\epsilon' A^{(i)}(S_j)}}$$

$$= \prod_{i=1}^{t} \frac{\sum_{j\geq i} e^{\epsilon' A^{(i)}(S'_j)}}{\sum_{j\geq i} e^{\epsilon' A^{(i)}(S_j)}},$$

where the last equality holds because if $i > t$, then the element $e_0$ was fully covered by iteration $i$, implying that $A^{(i)}(S_j) = A^{(i)}(S'_j)$ for all $j \geq i$. Also, given that all elements are eventually covered, we have $\sum_{i=1}^{n} A^{(i)}(S_i) = \sum_{i=1}^{n} A^{(i)}(S'_i)$.

Assuming $k \leq t$, we break up the product $\prod_{i=1}^{t} \frac{\sum_{j\geq i} e^{\epsilon' A^{(i)}(S'_j)}}{\sum_{j\geq i} e^{\epsilon' A^{(i)}(S_j)}}$ into three terms $I_1, I_2, I_3$ as follows:

$$I_1 \cdot I_2 \cdot I_3 := \left(\prod_{i=1}^{k-1} \frac{\sum_{j\geq i} e^{\epsilon' A^{(i)}(S'_j)}}{\sum_{j\geq i} e^{\epsilon' A^{(i)}(S_j)}}\right) \cdot \left(\frac{\sum_{j\geq k} e^{\epsilon' A_t(S'_j)}}{\sum_{j\geq k} e^{\epsilon' A_t(S_j)}}\right) \cdot$$

$$\left(\prod_{i=k+1}^{t} \frac{\sum_{j\geq i} e^{\epsilon' A^{(i)}(S'_j)}}{\sum_{j\geq i} e^{\epsilon' A^{(i)}(S_j)}}\right).$$

In the case when $t < k$ (or $t \leq k$) the terms $I_2$ and $I_3$ (or just $I_3$) vanish, and $k$ is replaced by $t$. However, the argument by enlarge would remain unaffected by this adjustment.

We proceed by considering two possible cases: $m(S_k, e_0) > m(S'_k, e_0)$ and $m(S_k, e_0) < m(S'_k, e_0)$.

$\mathbf{m(S_k, e_0) > m(S'_k, e_0)}$. In this scenario, both $I_1$ and $I_2$ are less than or equal to 1. Therefore, it is sufficient to focus on upper bounding $I_3$. Define an index-set

$$S^{I,i} := \{j : A^{(i)}(S_j) \neq A^{(i)}(S'_j)\}.$$

Note that $r^{(i-1)}(e_0) < r^{(i-1)}(e'_0)$ when $k < i < t$, and thus, $A^{(i)}(S) \leq A^{(i)}(S')$ for $k < i \leq t$. Specifically, we have that $A^{(i)}(S'_j) = A^{(i)}(S_j) + 1$ for any $j \in S^{I,i}$. Therefore, we can write $I_3$ as

$$I_3 = \prod_{i=k+1}^{t} \frac{\sum_{j\geq i} e^{\epsilon' A^{(i)}(S'_j)}}{\sum_{j\geq i} e^{\epsilon' A^{(i)}(S_j)}} = \prod_{i=k+1}^{t} \frac{(e^{\epsilon'} - 1)\sum_{j\in S^{I,i}} e^{\epsilon' A^{(i)}(S_j)} + \sum_{j\geq i} e^{\epsilon' A^{(i)}(S_j)}}{\sum_{j\geq i} e^{\epsilon' A^{(i)}(S_j)}}$$

$$= \prod_{i=k+1}^{t} \left(1 + (e^{\epsilon'} - 1) \cdot \frac{\sum_{j\in S^{I,i}} e^{\epsilon' A^{(i)}(S_j)}}{\sum_{j\geq i} e^{\epsilon' A^{(i)}(S_j)}}\right).$$

Recall that $\Pr[\pi(i) = j] = \frac{e^{\epsilon' A^{(i)}(S_j)}}{\sum_{j \geq i} e^{\epsilon' A^{(i)}(S_j)}}$ is probability of sampling set $S_j$ at epoch $i$. Therefore, we can write the above expression simply as

$$\prod_{i=k+1}^{t} \left(1 + (e^{\epsilon'} - 1) \cdot \Pr[\pi(i) \in S^{I,i}]\right) = \prod_{i=k+1}^{t} (1 + (e^{\epsilon'} - 1) \cdot \Pr[S^{I,i}]).$$

Additionally, observe that to sample a set from $S^{I,i}$ means to fully cover $e_0$, which can only occur at step $t$. Next, we will need the following lemma from [19]:

**Lemma C.3.** *The probabilistic process is modeled by flipping a coin over $t$ rounds. $p_i$ is the probability that it would come up heads in round $i$, and $p_i$ can be chosen adversarially based on the previous $i - 1$ rounds. Let $Z_i$ be the indicator for the event that no coin comes up heads in the first $i$ steps. Let $Y = \sum_{i=1}^{t} p_i Z_i$. Then for any $q$, $\Pr[Y > q] \leq \exp(-q)$.*

In our setup, $Z_i$ corresponds to the indicator of the event "$e_0$ is fully covered at round $i$". If $\sum_{i=k+1}^{t-1} \Pr[S^{I,i}] Z_i \leq \ln \delta^{-1}$, then we obtain

$$\frac{\Pr[M(\mathcal{A}) = \pi]}{\Pr[M(\mathcal{A}') = \pi]} \leq I_3$$

$$\leq \prod_{i=k+1}^{t} \exp((e^{\epsilon'} - 1) \Pr[S^{I,i}]) \qquad \text{since } 1 + x \leq e^x$$

$$\leq \exp(2\epsilon' \sum_{i=k+1}^{t} \Pr[S^{I,i}]) \qquad e^x \leq 1 + 2x \text{ for small } x \geq 0$$

$$\leq \exp(2\epsilon'(\ln(1/\delta) + \Pr[S^{I,t}]))$$

$$\leq \exp(2\epsilon'(\ln(1/\delta) + 1)) = \exp(\epsilon).$$

Finally, by Lemma C.3, the probability of the event $\sum_{i=k+1}^{t-1} \Pr[S^{I,i}] Z_i > \ln \delta^{-1}$ is upper bounded by $\delta$. Consequently, if $\mathcal{P}$ denotes the set of outcomes, we conclude that

$$\Pr[M(\mathcal{A}) \in \mathcal{P}] \leq \exp(\epsilon) \Pr[M(\mathcal{A}') \in \mathcal{P}] + \delta.$$

For a more detailed proof, see [19].

$\mathbf{m}(\mathbf{S_k}, \mathbf{e_0}) < \mathbf{m}(\mathbf{S'_k}, \mathbf{e_0})$. In this scenario, $I_3 \leq 1$. Our focus then shifts to $I_1 \cdot I_2$, following an analogous argument to the one discussed above, we obtain

$$I_1 \cdot I_2 = \prod_{i=1}^{k} \frac{\sum_{j \geq i} e^{\epsilon' A^{(i)}(S'_j)}}{\sum_{j \geq i} e^{\epsilon' A^{(i)}(S_j)}}$$

$$= \prod_{i=1}^{k} (1 + (e^{\epsilon'} - 1) \cdot \Pr[\pi(i) \in S^{I,i}])$$

$$= \prod_{i=1}^{k} (1 + (e^{\epsilon'} - 1) \cdot \Pr[\pi(i) = S_k]) \qquad \text{since } S^{I,i} = \{k\} \text{ for } i \leq k$$

$$\leq \prod_{i=1}^{k} \exp((e^{\epsilon'} - 1) \Pr[\pi(i) = S_k]) \qquad \text{since } 1 + x \leq e^x$$

$$\leq \exp(2\epsilon' \sum_{i=1}^{k} \Pr[\pi(i) = S_k]) \qquad e^x \leq 1 + 2x \text{ for small } x \geq 0$$

We then apply Lemma C.3 analogous to the above discussion, which completes the proof.

Next, we turn to the instance where the neighboring problems have different covering constraints $R \neq R'$. As before, let $t$ denote the epoch at which the covering constraint for $e_0$ is satisfied by both $M(\mathcal{A})$ and $M(\mathcal{A}')$. Although $\mathcal{S} = \mathcal{S}'$, we refer to the sets in $\mathcal{S}'$ as $S'$ for for clarity.

$\mathbf{r_{e_0}} > \mathbf{r'_{e_0}}$. This case is straightforward, since $\sum_{i=1}^{n} A^{(i)}(S_i) - \sum_{i=1}^{n} A^{(i)}(S'_i) = r_{e_0} - r'_{e_0} = 1$. We obtain

$$\frac{\Pr[M(\mathcal{A}) = \pi]}{\Pr[M(\mathcal{A}') = \pi]} = e^{\epsilon'} \prod_{i=1}^{t} \frac{\sum_{j \geq i} e^{\epsilon' A(S'_j)}}{\sum_{j \geq i} e^{\epsilon' A(S_j)}} \leq \exp(\epsilon').$$

$\mathbf{r_{e_0}} < \mathbf{r'_{e_0}}$. In this case we have

$$\frac{\Pr[M(\mathcal{A}) = \pi]}{\Pr[M(\mathcal{A}') = \pi]} = e^{-\epsilon'} \prod_{i=1}^{t} \frac{\sum_{j \geq i} e^{\epsilon' A(S'_j)}}{\sum_{j \geq i} e^{\epsilon' A(S_j)}}$$

$$\leq e^{-\epsilon'} \prod_{i=1}^{t} \left(1 + (e^{\epsilon'} - 1) \cdot \Pr[\pi(i) \in S^{I,i}]\right)$$

$$= e^{-\epsilon'} \prod_{i=k+1}^{t} (1 + (e^{\epsilon'} - 1) \cdot \Pr[S^{I,i}]).$$

The remainder of the proof utilizes the same arguments as previously discussed. □

**Lemma C.4.** *Algorithm 5 runs in $\tilde{O}(qf|\mathcal{S}|)$, where $q$ is the maximum set size and $f$ is the maximum frequency of any element (ignoring multiplicity).*

*Proof.* Initially, the algorithm computes $A^{(1)}(\cdot)$ for all sets in $\mathcal{S}$, which can be done in $O(|\mathcal{S}|q)$. Then the algorithm runs for $|\mathcal{S}|$ iterations, once per set, contributing the $|\mathcal{S}|$ factor. In each iteration, a set is sampled according to the exponential mechanism, where probabilities are proportional to $\exp(\varepsilon' A(S))$. Sampling can be done in $\tilde{O}(1)$.

After a set $S_j$ is selected, the algorithm updates the covering requirements for each element $e \in S_j$, which affects the utilities $A(S')$ for all other sets $S'$ containing $e$. Since each element appears in at most $f$ sets, and each set contains at most $q$ elements, the number of affected utilities per iteration is at most $qf$. □

### C.1.2   Weighted case.

Here we briefly discuss the weighted version of PRIVATEMULSET, and adapt the methodology of [19] with some minor modifications. First, we may assume without loss of generality that $\min_{S \in \mathcal{S}} C(S) = 1$, and $W = \max_{S \in \mathcal{S}} C(S)$ with $n = |\mathcal{S}|$. Let $M = \sum_{e \in U} r_e$. Similar to the unweighted version, we define $A(S) = \sum_{e \in S} \min(r_e, m(S, e))$ for a set $S \in \mathcal{S}$, and we say that the *utility* $u(S)$ is defined to be equal to $A(S) - C(S)$. Additionally, we add a dummy set halve to $\mathcal{S}$ with utility $u(\text{halve}) = -T$ for $T = \Theta(\frac{\log n + \log \log(MW)}{\epsilon'})$. When halve is selected by Algorithm 6, it indicates that no set was actually chosen. Additionally, unlike other selections, halve is never removed from $\mathcal{S}$.

**Lemma C.5.** *The cost of the output of 6 is at most $O(T \log n \cdot OPT)$ with probability at least $1 - 1/poly(n)$.*

*Proof.* This follows from a verbatim argument in [19] with $n$ replaced by $M$ and $m$ replaced by $n$ in our notation. □

**Lemma C.6.** *6 is $(\epsilon, \delta)$-differentially private.*

*Proof.* The proof is identical to the privacy proof of the algorithm in the unweighted case, with $A(S)$ replaced by $u(S)$. □

Identically to [19], we can remove the dependency on $W$ to obtain an $O(\log M(\log n + \log \log M/\epsilon))$-approximation.

**Algorithm 6** Private algorithm for WEIGHTEDPRIVATEMULSET

---

1: **Input:** $(\epsilon, \delta)$, set system $\mathcal{S}$, covering requirement $R = \{r_e\}_{e \in U}$
2: $\epsilon' \leftarrow \frac{\epsilon}{2\ln(e/\delta)}$, initialize permutation $\pi \leftarrow \emptyset$
3: $\theta \leftarrow M, T = \Theta\left(\frac{\log n + \log\log(MW)}{\epsilon'}\right)$
4: $i \leftarrow 1, r_e^{(0)} \leftarrow r_e$ for all $e \in U, \mathcal{S}^{(1)} \leftarrow \mathcal{S}$
5: **while** $\theta \geq 1/W$ **do**
6:     Define $u^{(i)}(S) := \sum_{e \in S} \min(m(S, e), r_e^{(i-1)}) - C(S)/\theta$
7:     Sample $S \in \mathcal{S}^{(i)}$ with probability $\propto \exp(\epsilon' u^{(i)}(S))$
8:     **if** $S = \mathtt{hal}$ **then**
9:         $\theta \leftarrow \theta/2$
10:         $\mathcal{S}^{(i+1)} \leftarrow \mathcal{S}^{(i)}$
11:         $r_e^{(i)} \leftarrow r_e^{(i-1)}$ for all $e \in U$
12:     **else**
13:         Append $S$ to $\pi$
14:         $\mathcal{S}^{(i+1)} \leftarrow \mathcal{S}^{(i)} \setminus \{S\}$
15:         **for** $e \in S$ **do**
16:             $r_e^{(i)} \leftarrow \max(0, r_e^{(i-1)} - m(S, e))$
17:         **end for**
18:     **end if**
19: **end while**
20: **Output** $\pi$ concatenated with a random permutation of $\mathcal{S}^{(i)} \setminus \{\mathtt{hal}\}$

---

## C.2 MAXDEGREE

---

**Algorithm 1** (restated). Private algorithm for PRIVATEMAXDEG

---

1: **Input:** $(\epsilon, \delta)$, graph $G$, target degree $D$
2: Initialize set system $\mathcal{S} \leftarrow \emptyset$, requirements $R \leftarrow \emptyset$
3: **for** each $v \in V$ **do**
4:     Define multiset $S_v$ with $m(S_v, v) = \infty$ and $m(S_v, u) = 1$ for all $u \sim v$
5:     $\mathcal{S} \leftarrow \mathcal{S} \cup \{S_v\}, \quad R \leftarrow R \cup \{r_v = \max(\deg(v) - D, 0)\}$
6: **end for**
7: Set $\epsilon' \leftarrow \epsilon/4, \delta' \leftarrow \delta/4e^{3\epsilon'}$
8: **Return:** Algorithm 5$(\epsilon', \delta', \mathcal{S}, R)$ /*Applying the private multi-set algorithm*/

---

**Theorem 4.2.** *Algorithm 1 is $(\epsilon, \delta)$-differentially private, and the cost of its output is $\hat{B} < |OPT_{\text{MAXDEG}}| \cdot O((1 + 1/\epsilon') \ln |V|)$ with high probability.*

*Proof.* Since PRIVATEMAXDEG reduces to PRIVATEMULSET, the optimal solutions for both problems are equivalent. In addition, since Algorithm 5 outputs a $O(\ln m/\epsilon' + \ln q)$-approximation, and $m = |V|, q = 2\,\mathrm{GS}(G) \leq 2|V|$, we have $O(\ln m/\epsilon' + \ln q) \leq O((1 + 1/\epsilon') \ln |V|)$. $\qquad\square$

### C.2.1 Explicit solution for MAXDEGREE

---

**Algorithm 2** (restated). Explicit solution algorithm for PRIVATEMAXDEG

---

1: **Input:** Instance of PRIVATEMAXDEG and a permutation $\pi$ obtained from the exponential mechanism
2: $T' \leftarrow 6\ln n/\epsilon' - Lap(2/\epsilon_1)$
3: **for** $i = 1$ to $n$ **do**
4:     $\gamma_i \leftarrow L^{(i)} - Lap(4/\epsilon_1)$
5: **end for**
6: Let $k$ be the first index such that $\gamma_k \leq T'$
7: **Output:** $\{\pi(1), \ldots, \pi(k)\}$

---

**Theorem 4.3.** *The output $k$ of Algorithm 2 satisfies $\Delta(G - \cup_{i=1}^{k}\{\pi_i\}) \leq D + O(\log n/\epsilon')$ with high probability. In addition, $k = O(OPT \cdot \log n/\epsilon')$ with high probability.*

*Proof.* It is well established that the AboveThreshold algorithm is $(\alpha, \beta)$ accurate, i.e., $\Pr[|L^{(k)} - 6\ln n/\epsilon'| > \alpha] \leq \beta$, with

$$\alpha = \frac{8(\log n + \log(2/\beta))}{\epsilon_1}.$$

Then, for $\beta = 1/n$, we obtain $\alpha = \frac{16\ln n + 8\ln 2}{\epsilon} = O(\log n/\epsilon')$. Thus, $L^{(i)} \leq O(\log n/\epsilon')$ with high probability.

On the other hand, observe that if $\Delta(G - \cup_{i=1}^{k}\{\pi(i)\}) > D + x$, then there is a node $j$ that has degree at least $D + x$. The multi-set corresponding to this node would have size at least $x$, since removing this node would satisfy its covering requirement completely. Therefore, $x \leq L^{(k)}$, and hence, $\Delta(G - \{\pi(i) : 1 \leq i \leq k\}) \leq D + x$ with probability at least $1 - 1/n$.

Let $\hat{k}$ be the "true" stopping point $L^{(\hat{k})} \leq 6\ln n/\epsilon'$. Using the proof for Lemma C.1, the exponential mechanism satisfies $\hat{k} \leq O(OPT \cdot \log n/\epsilon')$ when $L^{(i)} \geq 6\log n/\epsilon'$. It is sufficient to show that $k - \hat{k} \leq O(\log n/\epsilon)$. Observe that for $i \geq \hat{k}$, $L^{(i)} \leq 6\ln n/\epsilon'$ but $\gamma_i \geq T'$, the Laplace noise added to $L^{(i)}$ is greater than that added to $T$, this occurs with probability at most $1/2$ (since the Laplace distribution is symmetric about 0). Then the probability $\Pr[k - \hat{k} \geq \log_2 n] \leq 1/n$, so $k \leq O(OPT \cdot \log n/\epsilon')$ with high probability. $\qquad\square$

## C.3 Lower Bounds

In this section, we state the lower bounds of even outputting an explicit partial coverage requirements, that (1) any $(\epsilon, \delta)$-differentially private algorithm outputting (explicitly) a multiplicative coverage requirements (covers at least $\alpha r_e$ for all $e, \alpha < 1$) must output at least $m - 1$ sets, and (2) any $(\epsilon, \delta)$-differentially private algorithm outputting (explicitly) an additive coverage requirements (covers more than $r_e - \beta$ for all $e$) using no more than $O(\log n) + |OPT|$, where $OPT$ indicates the optimal solution without privacy, must do so with $\beta = \tilde{\Omega}(\log n)$. The multiplicative case is straightforward to verify, as setting $r_e = 1$ for some element $e$. Any multiplicative partial cover must cover at least a total copy of $e$. This impossibility of this instance is reduced to the impossibility of the set cover problem as stated by [19].

**Theorem C.7.** *Any $(\epsilon, \delta)$-differentially private algorithm outputting an additive coverage requirements explicitly (covers at least $r_e - \beta$ for all $e$) using less than $O(\log n) + |OPT|$ with probability at least $1 - C, C = n^{-\Omega(1)}$ must do so with $\beta = \tilde{\Omega}(\log n)$.*

*Proof.* Assume an algorithm $M$ that is $(\epsilon, \delta)$-DP with $\delta = O(1/poly(n))$ that outputs an explicit cover that can partially cover at least $r_e - \beta$, using less than $O(\log n) + OPT$ sets for all $e$ with probability $\Theta(1)$.

Let $\mathcal{U} = \{e\}$.

Let $\alpha$ be a positive constant, such that the number of sets that $M$ outputs no more $|OPT| + \alpha \log n$ with probability at least $1 - C$. Let $r_e^0 > \beta + 3\alpha \log n)$.

Let $S_1 = \{e \times (r_e^0 - \beta - \alpha \log n)\}$, i.e., a set with $(r_e^0 - \beta - \alpha \log n)$ copies of $e$. Let the set system be $\mathcal{S} = \{S_1, \{e\} \times (\beta + \alpha \log n)\}$.

Consider four instances of the the input with coverage requirements $I_1 = (r_e = r_e^0, \mathcal{S}), I_2 = (r_e = r_e^0 - \beta, \mathcal{S}), I_3 = (r_e = r_e^0 - \beta - 1, \mathcal{S}), I_4 = (r_e = r_e^0 - 1, \mathcal{S})$ respectively, with the set system $\mathcal{S}$. It is clear that $\mathcal{S}$ is enough to fully cover all the instances.

Let $S^* = \{S \subset \mathcal{S} : S_1 \in S, |S| \leq \alpha \log n\}$. In other words, each $S$ contains $S_1$ and up to $\alpha - 1$ copies of $\{e\}$. Then every $S \in S^*$ covers at most $r_e^0 - \beta - 1$ copies of $e$.

Consider instance $I_1$. $M$ guarantees to cover at least $r_e^0 - \beta$ copies of $e$ with probability at least $1 - C$. Therefore, $\Pr[M(I_1) \in S^*] \leq C$.

Consider instance $I_2$. Without privacy, $S_1$ is the optimal solution, hence $|OPT = 1|$ for $I_2$. If $S_1 \notin M(I_2)$, $M(I_2)$ must use at least $r_1^0 - \beta - \alpha \log n > \alpha \log n$ sets. With probability at least $1 - C$, the output contains $S_1$ and using no more than $\alpha \log n$ sets, hence $\Pr[M(I_2) \in S^*] \geq 1 - C$.

Because $I_1, I_2$ are $\beta$-step neighbors, using group privacy we have:

$$1 - C \leq \Pr[M(I_2) \in S^*]$$
$$\leq e^{\beta \epsilon} \Pr[M(I_1) \in S^*] + \beta e^{\beta \epsilon} \delta$$
$$\leq e^{\beta \epsilon} C + \beta e^{\beta \epsilon} \delta.$$

Therefore $\frac{1 - e^{\beta \epsilon} C}{\beta e^{\beta \epsilon}} \leq 1/poly(n)$. It is clear that $\beta = \tilde{\Omega}(\log n)$.

$\square$

**Lemma 4.4. Lower bound of PRIVATEMAXDEG.** *Any explicit $(\epsilon, \delta)$-differentially private algorithm for the PRIVATEMAXDEGREE removing at most $O(\log n) + |OPT|$ nodes with probability at least $1 - C, C = n^{-\Omega(1)}$, must incur an additive error $\Delta(G - \cup_{i=1}^{k}\{\pi_i\}) = D + \tilde{\Omega}(\log n)$, where $\pi(1), \ldots, \pi(k)$ are the removed nodes.*

Using the same setup as in Theorem C.7, setting $r_v = \max(d(v, G) - D, 0)$ for all nodes $v$. Similar to Theorem C.7, any explicit $(\epsilon, \delta)$-DP algorithm removing fewer than $O(\log n) + |OPT|$ nodes will guarantee to cover each node $v$ no more than $d(v, G) - D - \tilde{\Omega}(\log n)$ times, i.e., the maximum degree of the remaining graph is $\Delta - (\Delta - D - \tilde{\Omega}(\log n)) = D + \tilde{\Omega}(\log n)$.

## D SPECTRALRADIUS

### D.1 Bound via PARTIALSETCOVER

---

**Algorithm 3** (restated). Private Hitting Walks Algorithm for PRIVMINSR

---

1: **Input:** Graph $G = (V, E)$, privacy parameters $(\epsilon, \delta)$
2: Set $\epsilon' \leftarrow \epsilon/(2\ln(e/\delta))$, initialize permutation $\pi \leftarrow \emptyset$
3: **for** $i = 1$ to $n$ **do**
4:     Sample $v \in V$ with prob. $\propto \exp(\epsilon' \cdot A(v))$,     append $v$ to $\pi$ and remove $v$ from $V$
5: **end for**
6: Set $T \leftarrow \Delta^{1/2}, \theta \leftarrow 4nT^4, \hat{\theta} \leftarrow \theta - Lap(2/\epsilon_1)$
7: **for** $i = 1$ to $n$ **do**
8:     $\gamma_i \leftarrow W_4(G[V - \{\pi(1), \ldots, \pi(i)\}]) - Lap(4/\epsilon_1)$
9: **end for**
10: Let $k$ be the first iteration such that $\gamma_k \leq \hat{\theta}$
11: **Output:** $(\pi(1), \ldots, \pi(k))$

---

**Lemma D.1.** *If $T^4 \geq 6\ln n/\epsilon'$, the output $V' = \{\pi_1, \ldots, \pi_k\}$ of Algorithm 3 satisfies $W_4(G[V \setminus V']) \leq nT^4 + O(\log n/\epsilon')$ and gives an $O(\log n)$ approximation with high probability; the algorithm is $(\Delta^2(\epsilon + \epsilon_1), \Delta^2 \delta e^{(\Delta^2 - 1)\epsilon})$-differentially private and runs in time $\tilde{O}(n\Delta^4 \omega^4)$, where $\omega$ is the matrix multiplication exponent for $n \times n$ matrices.*

*Proof.* Since AboveThreshold is $(\alpha, \beta)$-accurate, for $\beta = 1/n$, we obtain $W_4[V \setminus V'] \leq nT^4 + O(\log n/\epsilon')$ whp, similar to the proof for Theorem 4.3.

Let $L_i$ denote the utility of the largest set after the $V_i = \{\pi_1, \ldots, \pi_i\}$ have been removed (i.e. $L_i = \max_v A(v)$). For $i < k$, $W_4(V \setminus V_i) \geq nT^4 \geq n \cdot 6\ln n/\epsilon'$, and $W_4(V \setminus V_i) \leq \sum_{v \in V} A(v) \leq nL_i$. Hence, $L_i \geq 6\ln n/\epsilon'$. By the same argument as in Proof 4.3, $A(\pi_i) \geq L_i/2$ whp. In other words, the utility of the chosen set is at least half of that chosen by a non-private greedy algorithm. Since the greedy algorithm is a $O(\ln n)$ approximation, Algorithm 3 would be a $O(2\ln n) = O(\ln n)$-approximation. $\square$

**Lemma D.2.** *Algorithm 3 is* $(\Delta^2(\epsilon + \epsilon_1), \Delta^2 \delta e^{(\Delta^2-1)\epsilon})$-*private.*

*Proof.* Since $A(v)$ has a sensitivity of $\Delta^2$, neighboring datasets in Private Hitting Walks would be $\Delta^2$-step neighbors in Partial Set Cover and Above Threshold instead.

Algorithm 3 is the composition of a $(\Delta^2 \epsilon, \Delta^2 \delta e^{(\Delta^2-1)\epsilon})$-private set cover algorithm and $\Delta^2 \epsilon_1$-private AboveThreshold process, hence the overall privacy budget would be $(\Delta^2(\epsilon + \epsilon_1), \Delta^2 \delta e^{(\Delta^2-1)\epsilon})$. $\quad\square$

### D.2 PRIVATESPECTRALRADIUS via PRIVATEMULSET

---

**Algorithm 4** (restated). Private algorithm for PRIVATEMAXDEG

---

1: **Input:** $(\epsilon, \delta)$, input graph $G$, target max degree $D$
2: $\mathcal{S} = \{S_v : v \in V\}$, such that $S_v$ contains $\infty$ copies of $v$ and $d(u, G)$ copies of each $u$ that is adjacent to $v$
3: $R = \{r_V : v \in V\}, \forall v \in V : r_v \leftarrow \max(\sum_{u \sim v} d(u, G) - D, 0)$
4: $\epsilon' \leftarrow \epsilon/4\Delta$
5: $\delta' \leftarrow \delta/4\Delta e^{(4\Delta-1)\epsilon'}$
6: **Return** Algorithm $5(\epsilon', \delta', \mathcal{S}, R)$

---

We approach the problem of reducing $\max_{u \in V(G)} \sqrt{\sum_{v \sim u} d(v, G)}$ using a similar strategy as in the PRIVATEMAXDEG case – by reformulating it as an instance of PRIVATEMULSET. First, we define sets $\{S_u\}_{u \in V}$, so that $m(S_u, u) = \infty$ and $m(S_u, v) = d(u, G)$ for all vertices $v$ adjacent to $u$. Additionally, for each vertex $u \in V$, set $r_u = \max(0, \sum_{v \sim u} d(v, G) - D)$, where $\sqrt{D}$ is a target upper bound. We can then apply the same analysis used in the PRIVATEMAXDEG case. However, we must adjust our edge-privacy model for this scenario. In the worst case, adding an edge $(u, v)$ could cause neighboring graphs in the PRIVATEMULSET formulation to become $4\Delta$-neighbors. This happens because such an edge addition can increase both the covering requirements $r_u$ and $r_v$ as well as the multiplicities $m(S_u, v)$ and $m(S_v, u)$ by up to $\Delta$. Thus, the algorithmic approach and results from PRIVATEMAXDEG largely carry over, but the sensitivity needs to be adjusted from 4 to $4\Delta$. All combined we obtain the following result:

**Theorem 5.2.** *Algorithm 4 is* $(\epsilon, \delta)$-*differentially private, and the cost of its output is* $\hat{B} < |OPT| \cdot O((1 + 1/\epsilon') \ln |V|)$ *with high probability.*

## E  Additional Experiments

**Effect of the $\epsilon$ on the spectral radius.** Figure 4 shows the $\rho(G[V - S])$ for the explicit solutions $S$ computed using Algorithm 2 for BTER networks, for the same parameters and privacy budgets mentioned earlier. The results here show that the resulting spectral radius is quite a bit smaller than the maximum degree. As expected, the resulting spectral radius of the residual graphs follow a similar trend as the max degree, with higher $\epsilon$ budgets obtaining better metrics due to less privacy constraints.

**Cost of achieving different epidemic metrics.** Figures 5 and 6 show the violation in the target degree (for $D = 20$) and the spectral radius vs the explicit solution cost (computed using Algorithm 2), for different $\epsilon$ in the BTER networks. As noted earlier, the violation and spectral radius decrease significantly as the solution cost increases, which is achieved for higher $\epsilon$.

**Privacy vs Vaccination Cost in BTER.** We also investigated the tradeoff of privacy and vaccination cost in the 3 BTER graphs ($\gamma = 0.5, \rho = 0.95, \eta = 0.05$) for implicit PRIVATEMAXDEG, with target degree $D = 20$, as shown in Figure 7. The non-private greedy algorithm is used as a baseline comparison. Due to the relaxed privacy budget of $\delta = 0.01$, the variation of $\epsilon$ has a much more pronounced effect on vaccination budget, and the algorithm's performance is much closer to that of the non-private greedy as compared to Figure 1, and are within the bounds expected from Lemma 4.2.

**Effect on Infection Simulation.** Finally, we computed the 300 explicit solutions using various privacy budgets $\epsilon$ (and $\delta = 0.01$) and target max degree 10 for 3 "social circles" in the SNAP Facebook datasets [29], we then performed 200 simulations of SIR with transmission probability 0.2 and 20 initial infections to determine the average vaccination budget and infection size and

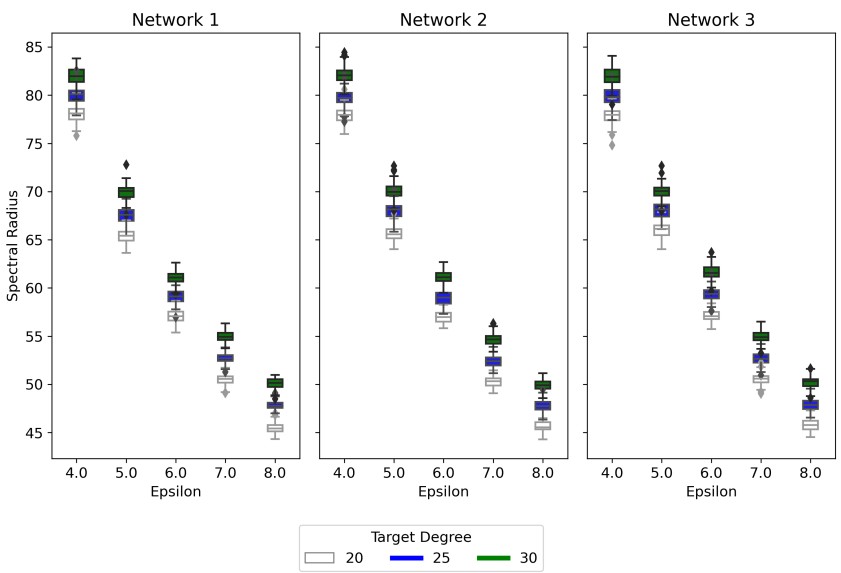

Figure 4: Effect of Privacy on Spectral Radius on BTER Graphs

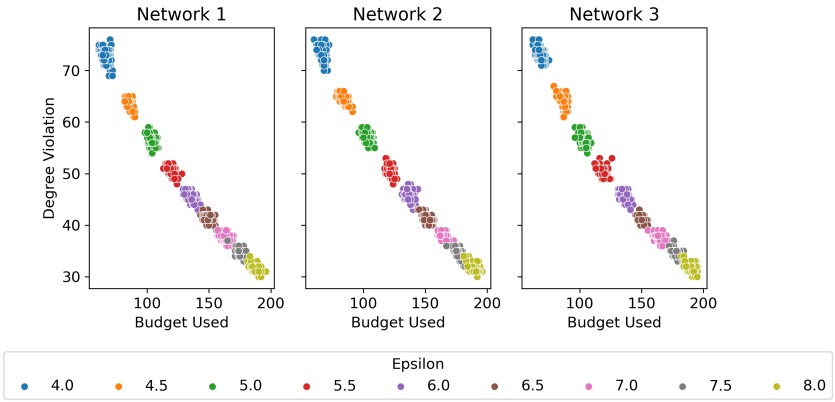

Figure 5: Tradeoff of Degree Violation vs Budget on BTER Graphs

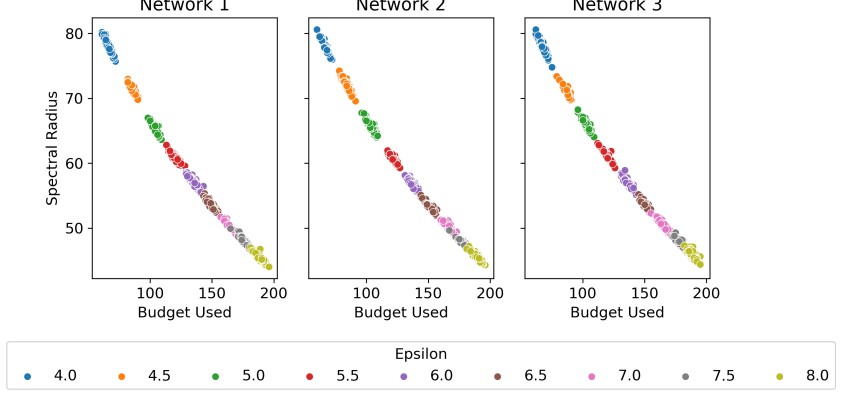

Figure 6: Tradeoff of Spectral Radius vs Budget on BTER Graphs

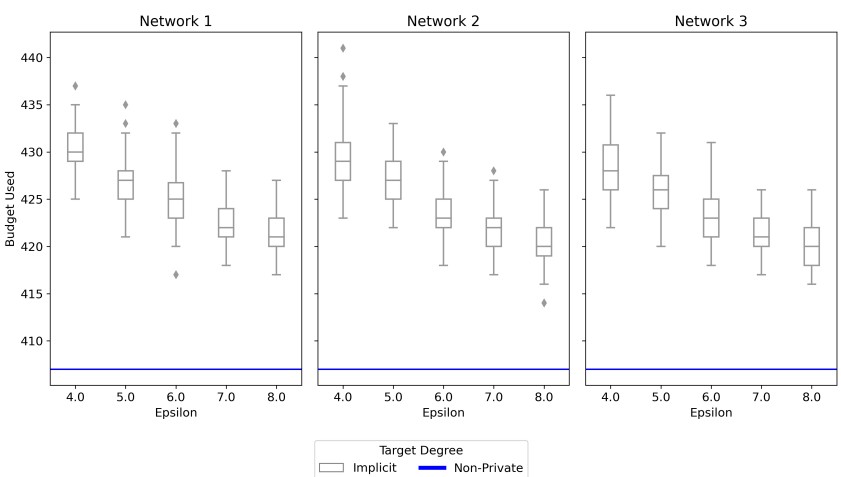

Figure 7: Tradeoff of Spectral Radius vs Budget on BTER Graphs for Implicit Solution ($D = 20$)

Table 3: Infection Spread on Facebook Social Circles

| Network | $\epsilon = 4$ | | $\epsilon = 6$ | | $\epsilon = 8$ | |
|---|---|---|---|---|---|---|
| | Budget | Spread | Budget | Spread | Budget | Spread |
| Circle 0 | 14.52 | 205.18 | 30.48 | 171.55 | 42.28 | 138.02 |
| Circle 1 | 311.70 | 586.99 | 411.53 | 413.50 | 546.56 | 251.49 |
| Circle 2 | 45.52 | 138.29 | 73.45 | 90.07 | 94.57 | 60.38 |

demonstrate the effectiveness of the solutions to minimize infection spread. Note that we used the more relaxed mutltiset multicover version of differential privacy for these experiments.

