# OpenReview forum: "Controlling The Spread of Epidemics on Networks with Differential Privacy"
_NeurIPS.cc/2025/Conference — NeurIPS 2025 poster_

### Official Review · Reviewer_Aeve · 2025-06-18

**Clarity:** 2
**Significance:** 3
**Originality:** 2
**Rating:** 5
**Confidence:** 4

**Summary:**

This paper gives a differentially private algorithm for two important problems.

(1) PrivateMaxDeg: Remove the minimum number of nodes such that the remaining graph has maximum degree no larger than a given value $D$.

(2) PrivMinSR: Remove the minimum number of nodes such that the remaining graph has largest eigenvalue no larger than a given value $\Delta$.

The authors give algorithms with theoretical guarantees, give lower bounds to demonstrate that the algorithms are tight, and also give experimental results for the proposed algorithms.

**Questions:**

I am willing to raise my score if the authors can clarify the derivation at Line 754 and confirm that it is not a typographical error or an omission in explanation.

**Ethical Concerns:**

["NO or VERY MINOR ethics concerns only"]

**Final Justification:**

I have confirmed that all the proofs are sound, and the paper has a clear contribution.

**Limitations:**

yes

**Quality:**

3

**Strengths And Weaknesses:**

I think that this paper has a very high potential. Unfortunately, I think that it may need some improvements on the following matters.

(1) The problems addressed in this paper are quite similar to graph coloring problems, and the techniques that the authors introduced are quite similar to theirs. I believe that the authors should have cited the following works and explain the differences between their techniques and the techniques introduced in those papers:

> Private graph colouring with limited defectiveness

> Aleksander B. G. Christiansen, Eva Rotenberg, Teresa Anna Steiner, Juliette Vlieghe

(2) I think the paper misses an explanation at the most important part of the proof that hinders me from understand the proof.
At line 754 of the full paper, I wonder why we can move from the second to the third, and also from the third to the fourth of the equation chain.

---

> ### Author Rebuttal · Authors · 2025-07-30
>
> We thank the reviewer for pointing out the lack of explanation in the proof in the appendix. We agree that this part of the argument can be difficult to follow, and we will revise the notation and provide additional clarification in the revised version. Below is an explanation of the equation chain:
>
> 1. Observe that for any pair of sets $S_j$ and $S_j'$, that are picked after step $t$, are identical, and $m(S_j, e_0) = m(S_j, e_0)$. However, $A(S_j)\leq A(S_j')$ since $e_0$ has been covered more in step $t$. The inequality is strict iff $S_{j'}$ fully satisfies $e_0$'s covering constraint of at the time it is selected.
> Therefore, by definition of $S^{I,i} = \{j:A_i(S_j)<A_i(S_j')\}$, for any $j\in S^{I,i}$ we have $A_i(S_j') = A_i(S_j)+1$, which implies the transition from line 1 to line 2.
>
> 2. For line 2 $\to$ line 3. Recall that in step $i$, set $S_{\pi_i}$, is sampled from among $j>t$ with probability proportional to $\exp(\epsilon'A_i(S_j))$, so it becomes $(\sum_{j\in S^{I,i}}Pr[j])/(\sum_{j\geq i}Pr[j])$, which corresponds to the probability of selecting an index $j \in S^{I,i}$ conditioned on $j \geq i$.
>
> 3. Line 3 $\to$ line 4. The numerator in the previous expression is simply the probability that $\pi_i\in S^{I,i}$, which we denote as $\Pr[S^{I,i}]$.

---

> ### Comment · Reviewer_Aeve · 2025-08-04
> **Thank you very much for the reply**
>
> I think I can now understand the proof. I therefore will increase my score.

---

### Official Review · Reviewer_MSr6 · 2025-07-08

**Clarity:** 3
**Significance:** 3
**Originality:** 3
**Rating:** 4
**Confidence:** 5

**Summary:**

This paper studies the problem of epidemic control on networks under edge-differential privacy. It focuses on designing vaccination strategies by minimizing network properties such as the maximum degree (PRIVATEMAXDEG) and the spectral radius (PRIVMINSR). The authors reduce these problems to a differentially private version of the multi-set multi-cover problem (PRIVATEMULSET), and develop both implicit and explicit algorithms using techniques like the exponential mechanism and AboveThreshold. Experimental results are provided on real and synthetic networks to evaluate the trade-off between privacy and utility under different privacy budgets.

**Questions:**

- Figure 2 shows an unexpected trend where stronger privacy (lower epsilon) leads to lower vaccination cost. The explanation is brief and needs more justification. The paper does not discuss how to choose an optimal epsilon, nor what happens when epsilon is very small (e.g., < 4) in Figure 2.

**Ethical Concerns:**

["NO or VERY MINOR ethics concerns only"]

**Final Justification:**

After considering the rebuttal, other reviews, and the AC’s comment, I am raising my score by 1. The authors clarified the scope of their theoretical guarantees and addressed applicability concerns. While some scalability and utility–privacy trade-off issues remain, these are reasonable for future work. Given the novelty, relevance, and positive consensus, I find acceptance reasonable.

**Limitations:**

YES

**Quality:**

2

**Strengths And Weaknesses:**

Strengths:
- The paper addresses an important and timely problem of epidemic control under differential privacy.
- It introduces a novel reduction of PRIVATEMAXDEG and PRIVMINSR to a private multi-set multi-cover problem (PRIVATEMULSET), enabling algorithmic design using known private mechanisms.

Weaknesses:
- The connection between PRIVMINSR and PRIVATEMULSET is not clearly demonstrated. A concrete example would help clarify the transformation.
- The use of different privacy parameters (epsilon, epsilon', epsilon_1, delta') across lemmas and algorithms is confusing and lacks clear explanation.
- In the Montgomery County subnet, the private algorithm requires a vaccination budget nearly 10 times higher than the non-private baseline, which raises practical concerns.

---

> ### Author Rebuttal · Authors · 2025-07-30
>
> We thank the reviewer for pointing this out — it is a rather unexpected result: Stronger privacy requires larger degree violations, which more than offsets the increased randomness in the permutations. However, the authors believe that the optimal choice of $\epsilon$ is best left to the governing body to determine as there are serious ethical implications.

---

> > ### Comment · Reviewer_MSr6 · 2025-08-06
> >
> > Thanks for the rebuttal. Finally, I choose to keep my score.

---

### Official Review · Reviewer_MnSu · 2025-07-14

**Clarity:** 4
**Significance:** 1
**Originality:** 3
**Rating:** 4
**Confidence:** 2

**Summary:**

This paper studies vaccination strategies for epidemic control on contact networks under differential privacy constraints. The authors focus on two key network properties known to influence epidemic spread: maximum degree and spectral radius. Their key technical approach reduces both problems to instances of a "multi-set multi-cover" problem, for which they design private algorithms using the exponential mechanism. They provide both implicit solutions (outputting a permutation from which nodes can determine inclusion) and explicit solutions (directly outputting the vaccination set).

**Questions:**

Mostly around significance and how applicable it is.

**Ethical Concerns:**

["NO or VERY MINOR ethics concerns only"]

**Limitations:**

More discussion on actual applicability. Their evaluation was on a small network, way smaller than a typical city where such a thing would be used.

**Quality:**

3

**Strengths And Weaknesses:**

Strength -
1. The lower bound results are impressive and establish the limitations in private cover problems.
2. The multi-set multi-cover reduction is clever and the approach to providing both implicit and explicit solutions addresses a known challenge in private covering problems.
3. Contact tracing privacy concerns were a major barrier during COVID-19, making this a practically motivated research direction even if the current results have limitations.

Weaknesses -
1. The edge-DP model assumes node identities are public while only edge relationships are private. Knowing who exists but not who they contact severely limits the model's applicability.
2. In covering problems, even the optimal non-private solution often requires vaccinating a significant fraction of nodes (hundreds to thousands). Multiplying this by large approximation factors makes the solution require vaccinating most of the network.
3. I think generally, the issue is that while logarithmic factors are asymptotically good, the hidden constants and practical parameter values make these bounds. Therefore, I recommend this paper to focus on the theory instead of tying the problem to actual application. If application is still their focus, then they have to first start with answering what is a practically acceptable privacy-utility trade-off and then make a convincing case whether their solution satisfies that.

---

### Official Review · Reviewer_7rGB · 2025-07-17

**Clarity:** 1
**Significance:** 4
**Originality:** 3
**Rating:** 4
**Confidence:** 4

**Summary:**

The paper studies the problem of designing interventions (vaccination or isolation strategies) to control the spread of epidemics through a network. The main challenge arises from having to do the interventions in a privacy-preserving way in order to protect sensitive information like which users share contact (edges of the network). The paper considers interventions that can influence network properties like Max Degree or Spectral Radius. Their main contribution is three-fold: i) developing a private algorithm for the general Multi-Set Multi-cover problem, ii) showing that the problems of minimizing max degree or spectral radius can be reduced to instances of the multi-set multi-cover problem which they already know how to solve privately, and iii) finally, demonstrating effectiveness of their algorithm on real-world and synthetic networks.

**Questions:**

Detailed Comments & Questions:

Section 4:

1. At the start of section 4.1, the authors say that they are discussing the Unweighted case, which according to their Defn 3.4 should try to minimize the number of sets to be chosen from S to achieve the desired coverage for all elements. However, their main result Lemma 4.1 mentions an approximation ratio on the worst case cost which leads the reviewer to think that this is the weighted case ? Can the authors clarify ?

2. What does f mean ? The statement says "maximum frequency of an element (ignoring multiplicity)" -- this is not clear. Are the authors talking about the maximum number of sets S_i any element can occur in ?

3. The authors should provide a description of their main algorithm, specifically including where the modifications are due to privacy. The entire description before Lemma 4.1 has no mention of privacy at all. They should at least mention that the sampling is done using the exponential algorithm on the total utility of set S_i.  This is especially important for readability because their other algorithms build on this.

4. The reviewer is not familiar with previous literature on the best achieved approximation ratio for the multi-set problem in the non-private case. Can the authors provide that and discuss how their ratio compares with it. It is expected to be worse due to differential privacy, but by how much?

5. I do not understand the distinction between implicit and explicit solutions in this case. In the normal setting, given a permutation \pi, we can map it back to a collection of sets {S_u} which can be mapped back to a set of nodes {u} to be treated/removed. Is the mapping hindered by privacy ? This should not be the case because the central planner (who is deciding where to intervene) knows the contact graph fully ? The impression I got from reading the introduction is that the goal is to design the intervention in a privacy-preserving way, but the designer knows the contact graph. Can the authors clarify the privacy model ?

6. I do not understand Lemma 4.5. Why does the privMaxDeg algorithm only achieve partial coverage ? My intuition from the privMultiSet algorithm is that privacy causes us to remove more sets than we actually need to, incurring a larger cost than OPT, but always achieving full coverage. If MaxDeg exactly reduces to MultiSet, why does full coverage break ?

7. Minor typo (Lemma 4.1): The authors probably mean O((ln m)/ϵ + ln q) · |OP T|, not ϵ'.

Section 5:
1. What is T in Sec 5.1 ?

2. Can the authors comment on the relationship between spectral radius and max degree of a graph? How should we interpret the results in Table 2 ? I ask this because Max Degree of a graph seems to be a much poorer indicator of how connected a graph is/how easily infection spreads over the network. Spectral Radius seems to be a more holistic metric to try and minimize. Can the authors provide examples of graph classes where Max Degree reduction is the better way to go ?


I will be willing to reconsider my score if my questions are answered meaningfully.

**Ethical Concerns:**

["NO or VERY MINOR ethics concerns only"]

**Final Justification:**

The authors engaged with the questions asked during review. Most of my technical concerns were addressed (including clarification on the privacy model, difference between the implicit and explicit solutions, other questions about notation). I have also read through other reviews to calibrate my judgement. I feel positively about the paper in general and am willing to look past some of the presentation issues highlighted during review. My final score is adjusted accordingly.

**Limitations:**

Overall, the reviewer feels that the paper would benefit greatly from a more streamlined way of presenting results. The paper in its current form is overloaded with technical results without (in many cases) clear explanation of why those results are important to the context. For example, why would a reader care about graph-theoretic connections between the spectral radius and walks of length 4. A lot of these unnecessary technicalities can be removed from the main body.  Lots of notation are introduced on the fly without formal definitions. Several results like runtime complexity can be omitted entirely from the main body or just mentioned in passing.

**Paper Formatting Concerns:**

None.

**Quality:**

3

**Strengths And Weaknesses:**

*Quality*: Although the reviewer has not checked the mathematical details/proofs, the paper appears to be technically thorough. Technical questions below.

*Significance and originality*: The paper studies a very important problem. Differential privacy has enormous scope in the public health domain. As the paper rightly points out, people are in general concerned about privacy which degrades the efficacy of interventions (for, example, reluctance to provide information to help with contact-tracing during Covid-19). From that point of view, the paper tries to make a meaningful contribution which is both significant and original.

*Clarity*: The paper has serious deficiencies when it comes to clarity. Please find detailed comments and questions below.

---

> ### Author Rebuttal · Authors · 2025-07-30
>
> We thank the reviewer for their thoughtful comments and useful suggestions, which we will incorporate in the revised version of the paper. We agree that including too many technical results in the main body limits clarity, and the paper would benefit from moving some of these details to the appendix. We will also revise the notation and streamline the presentation to improve overall readability. Below, we address the reviewer’s specific questions and clarify some notation:
>
> - Lemma 4.1 refers to the unweighted case, the cost $|OPT|$ is the number of sets required by the solution. $f$ is the maximum number of sets $S_i$ any element can occur in. This is clarified in the supplementary material, as this parameter is only used in the runtime analysis, which we plan to fully move to the appendix. The parameter $T$ in Section 5.1 is just a threshold value.
>
> - In the non-private setting, a greedy algorithm achieves an approximation ratio of $H(\max_S \sum_e m(S,e))$ (see [42]). While we mention this in the special case discussed in Section 6, we agree that introducing it earlier would help contextualize our approximation results and the trade-offs in the private setting.
>
> - The mapping from an implicit solution (a permutation) to a set of nodes is hindered by privacy, as it would depend on information about the edges, which are to be kept private. The central planner may conduct this step, at an additional privacy cost. We believe it inadvisable to publish the mapped set of nodes from the implicit solution on some special cases of PrivateMaxDeg: for example the case where target $D=0$ (vertex cover), in which every private solution must select at least all but one vertex.
>
> - Lemma 4.5 result is for the explicit solution algorithm, which does not guarantee full coverage of all elements. There are lower bounds to the cost of explicit private solutions. For instance, in the vertex cover case ($D=0$), a private algorithm must choose all but 1 vertex (otherwise, if there are two vertex not in the output, then one can conclude that there is no edge between them).
>
> - Indeed, minimizing solving MaxDegree problem provides weaker guarantees than solving spectral radius minimization. However, spectral radius is significantly harder to optimize, and the corresponding privacy and algorithm runtime guarantees are also weaker. In scenarios where privacy and efficiency are the primary concerns, using the max degree algorithms in Section 4 would be more practical.
>
> We appreciate the reviewer’s comments and believe that addressing these would help improve the clarity and accessibility of the paper.

---

> > ### Comment · Reviewer_7rGB · 2025-08-02
> > **Thanks for the rebuttal!**
> >
> > Thanks for the detailed response. I think, most of my questions have been answered. Overall, I think the paper is technically solid and makes good contributions to advancing theory of differential privacy on graph problems (despite some major presentation issues as highlighted during review). I hope, the authors will try to streamline the paper to make it more accessible to general audiences. I will update my score. Thanks!

---

### Decision · Program_Chairs · 2025-09-17

**Decision:**

Accept (poster)

**Comment:**

After the author response and discussion, the paper received unanimously positive scores. The review team is unanimous that the studied problem of deciding which nodes to protect under differential privacy constraints is interesting, novel, and timely, and the review team is generally happy with the state of the results. I will point out that there were concerns about the writing and clarity of the paper in places, that I would encourage the author team to address for the camera-ready.